# Orthogonally Decoupled Variational Gaussian Processes

**Hugh Salimbeni**\*
Imperial College London
hrs13@ic.ac.uk

**Ching-An Cheng**\*
Georgia Institute of Technology
cacheng@gatech.edu

**Byron Boots**
Georgia Institute of Technology
bboots@gatech.edu

**Marc Deisenroth**
Imperial College London
mpd37@ic.ac.uk

## Abstract

Gaussian processes (GPs) provide a powerful non-parametric framework for reasoning over functions. Despite appealing theory, its superlinear computational and memory complexities have presented a long-standing challenge. State-of-the-art sparse variational inference methods trade modeling accuracy against complexity. However, the complexities of these methods still scale superlinearly in the number of basis functions, implying that that sparse GP methods are able to learn from large datasets only when a small model is used. Recently, a decoupled approach was proposed that removes the unnecessary coupling between the complexities of modeling the mean and the covariance functions of a GP. It achieves a linear complexity in the number of mean parameters, so an expressive posterior mean function can be modeled. While promising, this approach suffers from optimization difficulties due to ill-conditioning and non-convexity. In this work, we propose an alternative decoupled parametrization. It adopts an orthogonal basis in the mean function to model the residues that cannot be learned by the standard coupled approach. Therefore, our method extends, rather than replaces, the coupled approach to achieve strictly better performance. This construction admits a straightforward natural gradient update rule, so the structure of the information manifold that is lost during decoupling can be leveraged to speed up learning. Empirically, our algorithm demonstrates significantly faster convergence in multiple experiments.

## 1 Introduction

Gaussian processes (GPs) are flexible Bayesian non-parametric models that have achieved state-of-the-art performance in a range of applications [8, 31]. A key advantage of GP models is that they have large representational capacity, while being robust to overfitting [27]. This property is especially important for robotic applications, where there may be an abundance of data in some parts of the space but a scarcity in others [7]. Unfortunately, exact inference in GPs scales cubically in computation and quadratically in memory with the size of the training set, and is only available in closed form for Gaussian likelihoods.

To learn from large datasets, variational inference provides a principled way to find tractable approximations to the true posterior. A common approach to approximate GP inference is to form a sparse variational posterior, which is designed by conditioning the prior process at a small set of *inducing points* [32]. The sparse variational framework trades accuracy against computation,

---

but its complexities still scale superlinearly in the number of inducing points. Consequently, the representation power of the approximate distribution is greatly limited.

Various attempts have been made to reduce the complexities in order to scale up GP models for better approximation. Most of them, however, rely on certain assumptions on the kernel structure and input dimension. In the extreme, Hartikainen and Särkkä [10] show that, for 1D-input problems, exact GP inference can be solved in linear time for kernels with finitely many non-zero derivatives. For low-dimensional inputs and stationary kernels, variational inference with structured kernel approximation [34] or Fourier features [13] has been proposed. Both approaches, nevertheless, scale exponentially with input dimension, except for the special case of sum-and-product kernels [9]. Approximate kernels have also been proposed as GP priors with low-rank structure [30, 25] or a sparse spectrum [19]. Another family of methods partitions the input space into subsets and performs prediction aggregation [33, 26, 24, 6], and Bayesian aggregation of local experts with attractive theoretical properties is recently proposed by Rullière et al. [28].

A recent decoupled framework [5] takes a different direction to address the complexity issue of GP inference. In contrast to the above approaches, this decoupled framework is agnostic to problem setups (e.g. likelihoods, kernels, and input dimensions) and extends the original sparse variational formulation [32]. The key idea is to represent the variational distribution in the reproducing kernel Hilbert space (RKHS) induced by the covariance function of the GP. The sparse variational posterior by Titsias [32] turns out to be equivalent to a particular parameterization in the RKHS, where the mean and covariance both share the same basis. Cheng and Boots [5] suggest to relax the requirement of basis sharing. Since the computation only scales linearly in the mean parameters, many more basis functions can be used for modeling the mean function to achieve higher accuracy in prediction.

However, the original decoupled basis [5] turns out to have optimization difficulties [11]. In particular, the non-convexity of the optimization problem means that a suboptimal solution may be found, leading to performance that is potentially worse than the standard coupled case. While Havasi et al. [11] suggest to use a pre-conditioner to amortize the problem, their algorithm incurs an additional cubic computational cost; therefore, its applicability is limited to small simple models.

Inspired by the success of natural gradients in variational inference [15, 29], we propose a novel RKHS parameterization of decoupled GPs that admits efficient natural gradient computation. We decompose the mean parametrization into a part that shares the basis with the covariance, and an orthogonal part that models the residues that the standard coupled approach fails to capture. We show that, with this particular choice, the natural gradient update rules further *decouple* into the natural gradient descent of the coupled part and the functional gradient descent of the residual part. Based on these insights, we propose an efficient optimization algorithm that preserves the desired properties of decoupled GPs and converges faster than the original formulation [5].

We demonstrate that our basis is more effective than the original decoupled formulation on a range of classification and regression tasks. We show that the natural gradient updates improve convergence considerably and can lead to much better performance in practice. Crucially, we show also that our basis is more effective than the standard coupled basis for a fixed computational budget.

## 2  Background

We consider the inference problem of GP models. Given a dataset $\mathcal{D} = \{(x_n, y_n)\}_{n=1}^N$ and a GP prior on a latent function $f$, the goal is to infer the (approximate) posterior of $f(x^*)$ for any query input $x^*$. In this work, we adopt the recent decoupled RKHS reformulation of variational inference [5], and, without loss of generality, we will assume $f$ is a scalar function. For notation, we use boldface to distinguish finite-dimensional vectors and matrices that are used in computation from scalar and abstract mathematical objects.

### 2.1  Gaussian Processes and their RKHS Representation

We first review the primal and dual representations of GPs, which form the foundation of the RKHS reformulation. Let $\mathcal{X} \subseteq \mathbb{R}^d$ be the domain of the latent function. A GP is a distribution of functions, which is described by a mean function $m : \mathcal{X} \rightarrow \mathbb{R}$ and a covariance function $k : \mathcal{X} \times \mathcal{X} \rightarrow \mathbb{R}$. We say a latent function $f$ is distributed according to $\mathcal{GP}(m, k)$, if for any $x, x' \in \mathcal{X}$, $\mathbb{E}[f(x)] = m(x)$, $\mathbb{C}[f(x), f(x')] = k(x, x')$, and for any finite subset $\{f(x_n) : x_n \in \mathcal{X}\}_{n=1}^N$ is Gaussian distributed.

We call the above definition, in terms of the *function values* $m(x)$ and $k(x, x')$, the *primal* representation of GPs. Alternatively, one can adopt a *dual* representation of GPs, by treating *functions* $m$ and $k$ as RKHS objects [4]. This is based on observing that the covariance function $k$ satisfies the definition of positive semi-definite functions, so $k$ can also be viewed as a reproducing kernel [3]. Specifically, given $\mathcal{GP}(m, k)$, without loss of generality, we can find an RKHS $\mathcal{H}$ such that

$$m(x) = \phi(x)^\top \mu, \qquad k(x, x') = \phi(x)^\top \Sigma \phi(x') \tag{1}$$

for some $\mu \in \mathcal{H}$, bounded positive semidefinite self-adjoint operator $\Sigma : \mathcal{H} \to \mathcal{H}$, and feature map $\phi : \mathcal{X} \to \mathcal{H}$. Here we use $^\top$ to denote the inner product in $\mathcal{H}$, even when $\dim \mathcal{H}$ is infinite. For notational clarity, we use symbols $m$ and $k$ (or $s$) to denote the mean and covariance functions, and symbols $\mu$ and $\Sigma$ to denote the RKHS objects; we use $s$ to distinguish the (approximate) posterior covariance function from the prior covariance function $k$. If $f \sim \mathcal{GP}(m, k)$ satisfying (1), we also write $f \sim \mathcal{GP}_\mathcal{H}(\mu, \Sigma)$.[*]

To concretely illustrate the primal-dual connection, we consider the GP regression problem. Suppose $f \sim \mathcal{GP}(0, k)$ in prior and $y_n = f(x_n) + \epsilon_n$, where $\epsilon_n \sim \mathcal{N}(\epsilon_n | 0, \sigma^2)$. Let $X = \{x_n\}_{n=1}^N$ and $\mathbf{y} = (y_n)_{n=1}^N \in \mathbb{R}^N$, where the notation $(\cdot)_{n=\cdot}$ denotes stacking the elements. Then, with $\mathcal{D}$ observed, it can be shown that $f \sim \mathcal{GP}(m, s)$ where

$$m(x) = \mathbf{k}_{x,X}(\mathbf{K}_X + \sigma^2 \mathbf{I})^{-1}\mathbf{y}, \qquad s(x, x') = k_{x,x'} - \mathbf{k}_{x,X}(\mathbf{K}_X + \sigma^2 \mathbf{I})^{-1}\mathbf{k}_{X,x'}, \tag{2}$$

where $k_{\cdot,\cdot}$, $\mathbf{k}_{\cdot,\cdot}$ and $\mathbf{K}_{\cdot,\cdot}$ denote the covariances between the subscripted sets,[†] We can also equivalently write the posterior GP in (2) in its dual RKHS representation: suppose the feature map $\phi$ is selected such that $k(x, x') = \phi(x)^\top \phi(x')$, then *a priori* $f \sim \mathcal{GP}_\mathcal{H}(0, I)$ and *a posteriori* $f \sim \mathcal{GP}_\mathcal{H}(\mu, \Sigma)$,

$$\mu = \Phi_X(\mathbf{K}_X + \sigma^2 \mathbf{I})^{-1}\mathbf{y}, \qquad \Sigma = I - \Phi_X(\mathbf{K}_X + \sigma^2 \mathbf{I})^{-1}\Phi_X^\top, \tag{3}$$

where $\Phi_X = [\phi(x_1), \ldots, \phi(x_N)]$.

## 2.2 Variational Inference Problem

Inference in GP models is challenging because the closed-form expressions in (2) have computational complexity that is cubic in the size of the training dataset, and are only applicable for Gaussian likelihoods. For non-Gaussian likelihoods (e.g. classification) or for large datasets (i.e. more than 10,000 data points), we must adopt approximate inference.

Variational inference provides a principled approach to search for an approximate but tractable posterior. It seeks a variational posterior $q$ that is close to the true posterior $p(f|\mathcal{D})$ in terms of KL divergence, i.e. it solves $\min_q \text{KL}(q(f)||p(f|\mathcal{D}))$. For GP models, the variational posterior must be defined over the entire function, so a natural choice is to use another GP. This choice is also motivated by the fact that the exact posterior is a GP in the case of a Gaussian likelihood as shown in (2). Using the results from Section 2.1, we can represent this posterior process via a mean and a covariance function or, equivalently, through their associated RKHS objects.

We denote these RKHS objects as $\mu$ and $\Sigma$, which uniquely determine the GP posterior $\mathcal{GP}_\mathcal{H}(\mu, \Sigma)$. In the following, without loss of generality, we shall assume that the prior GP is zero-mean and the RKHS is selected such that $f \sim \mathcal{GP}_\mathcal{H}(0, I)$ *a priori*.

The variational inference problem in GP models leads to the optimization problem

$$\min_{q=\mathcal{GP}_\mathcal{H}(\mu, \Sigma)} \mathcal{L}(q), \qquad \mathcal{L}(q) = -\sum_{n=1}^N \mathbb{E}_{q(f(x_n))}[\log p(y_n|f(x_n))] + \text{KL}(q(f)||p(f)), \tag{4}$$

where $p(f) = \mathcal{GP}_\mathcal{H}(0, I)$ and $\text{KL}(q(f)||p(f)) = \int \log \frac{q(f)}{p(f)} \mathrm{d}q(f)$ is the KL divergence between the approximate posterior GP $q(f)$ and the prior GP $p(f)$. It can be shown that $\mathcal{L}(q) = \text{KL}(q(f)||p(f|\mathcal{D}))$ up to an additive constant [5].

---

[*]This notation only denotes that $m$ and $k$ can be represented as RKHS objects, not that the sampled functions of $\mathcal{GP}(m, k)$ necessarily reside in $\mathcal{H}$ (which only holds for the special when $\Sigma$ has finite trace).

[†]If the two sets are the same, only one is listed.

## 2.3 Decoupled Gaussian Processes

Directly optimizing the possibly infinite-dimensional RKHS objects $\mu$ and $\Sigma$ is computationally intractable except for the special case of a Gaussian likelihood and a small training set size $N$. Therefore, in practice, we need to impose a certain sparse structure on $\mu$ and $\Sigma$. Inspired by the functional form of the exact solution in (3), Cheng and Boots [5] propose to model the approximate posterior GP in the *decoupled subspace parametrization* (which we will refer to as *decoupled basis* for short) with

$$\mu = \Psi_\alpha \mathbf{a}, \qquad \Sigma = I + \Psi_\beta \mathbf{A} \Psi_\beta^\top \qquad (5)$$

where $\alpha$ and $\beta$ are the sets of *inducing points* to specify the bases $\Psi_\alpha$ and $\Psi_\beta$ in the RKHS, and $\mathbf{a} \in \mathbb{R}^{|\alpha|}$ and $\mathbf{A} \in \mathbb{R}^{|\beta| \times |\beta|}$ are the coefficients such that $\Sigma \succeq 0$. With only finite perturbations from the prior, the construction in (5) ensures the KL divergence $\mathrm{KL}(q(f)||p(f))$ is finite [23, 5] (see Appendix A). Importantly, this parameterization decouples the variational parameters $(\mathbf{a}, \alpha)$ for the mean $\mu$ and the variational parameters $(\mathbf{A}, \beta)$ for the covariance $\Sigma$. As a result, the computation complexities related to the two parts become independent, and a *large* set of parameters can adopted for the mean to model complicated functions, as discussed below.

**Coupled Basis**   The form in (5) covers the sparse variational posterior [32]. Let $Z = \{z_n \in \mathcal{X}\}_{n=1}^M$ be some fictitious inputs and let $\mathbf{f}_Z = (f(z_n))_{n=1}^M$ be the vector of function values. Based on the primal viewpoint of GPs, Titsias [32] constructs the variational posterior as the posterior GP conditioned on $Z$ with marginal $q(\mathbf{f}_Z) = \mathcal{N}(\mathbf{f}_Z|\mathbf{m}, \mathbf{S})$, where $\mathbf{m} \in \mathbb{R}^M$ and $\mathbf{S} \succeq 0 \in \mathbb{R}^{M \times M}$. The elements in $Z$ along with $\mathbf{m}$ and $\mathbf{S}$ are the variational parameters to optimize. The mean and covariance functions of this process $\mathcal{GP}(m, s)$ are

$$m(x) = \mathbf{k}_{x,Z} \mathbf{K}_Z^{-1} \mathbf{m}, \qquad s(x, x') = k_{x,x'} + \mathbf{k}_{x,Z} \mathbf{K}_Z^{-1} (\mathbf{S} - \mathbf{K}_Z) \mathbf{K}_Z^{-1} \mathbf{k}_{Z,x}, \qquad (6)$$

which is reminiscent of the exact result in (2). Equivalently, it has the dual representation

$$\mu = \Psi_Z \mathbf{K}_Z^{-1} \mathbf{m}, \qquad \Sigma = I + \Psi_Z \mathbf{K}_Z^{-1} (\mathbf{S} - \mathbf{K}_Z) \mathbf{K}_Z^{-1} \Psi_Z^\top, \qquad (7)$$

which conforms with the form in (5). The computational complexity of using the coupled basis reduces from $O(N^3)$ to $O(M^3 + M^2 N)$. Therefore, when $M \ll N$ is selected, the GP can be applied to learning from large datasets [32].

**Inversely Parametrized Decoupled Basis**   Directly parameterizing the dual representation in (5) admits more flexibility than the primal function-valued perspective. To ensure that the covariance of the posterior strictly decreases compared with the prior, Cheng and Boots [5] propose a decoupled basis with an inversely parametrized covariance operator

$$\mu = \Psi_\alpha \mathbf{a}, \qquad \Sigma = (I + \Psi_\beta \mathbf{B} \Psi_\beta^\top)^{-1}, \qquad (8)$$

where $\mathbf{B} \succeq 0 \in \mathbb{R}^{|\beta| \times |\beta|}$ and is further parametrized by its Cholesky factors in implementation. It can be shown that the choice in (8) is equivalent to setting $\mathbf{A} = -\mathbf{K}_\beta^{-1} + (\mathbf{K}_\beta \mathbf{B} \mathbf{K}_\beta + \mathbf{K}_\beta)^{-1}$ in (5). In this parameterization, because the bases for the mean and the covariance are decoupled, the computational complexity of solving (4) with the decoupled basis in (8) becomes $O(|\alpha| + |\beta|^3)$, as opposed to $O(M^3)$ of (7). Therefore, while it is usually assumed that $|\beta|$ is in the order of $M$, with a decoupled basis, we can freely choose $|\alpha| \gg |\beta|$ for modeling complex mean functions accurately.

## 3   Orthogonally Decoupled Variational Gaussian Processes

While the particular decoupled basis in (8) is more expressive, its optimization problem is ill-conditioned and non-convex, and empirically slow convergence has been observed [11]. To improve the speed of learning decoupled models, we consider the use of natural gradient descent [2]. In particular, we are interested in the update rule for *natural parameters*, which has empirically demonstrated impressive convergence performance over other choices of parametrizations [29]

However, it is unclear what the natural parameters (5) for the general decoupled basis in (5) are and whether finite-dimensional natural parameters even exist for such a model. In this paper, we show that when a decoupled basis is appropriately structured, then natural parameters do exist. Moreover, they admit a very efficient (approximate) natural gradient update rule as detailed in Section 3.4. As a result, large-scale decoupled models can be quickly learned, joining the fast convergence property from the coupled approach [12] and the flexibility of the decoupled approach [5].

## 3.1 Alternative Decoupled Bases

To motivate the proposed approach, let us first introduce some alternative decoupled bases for improving optimization properties (8) and discuss their limitations. The inversely parameterized decoupled basis (8) is likely to have different optimization properties from the standard coupled basis (7), due to the inversion in its covariance parameterization. To avoid these potential difficulties, we reparameterize the covariance of (8) as the one in (7) and consider instead the basis

$$\mu = \Psi_\alpha \mathbf{a}, \qquad \Sigma = (I - \Psi_\beta \mathbf{K}_\beta^{-1} \Psi_\beta^\top) + \Psi_\beta \mathbf{K}_\beta^{-1} \mathbf{S} \mathbf{K}_\beta^{-1} \Psi_\beta^\top. \tag{9}$$

The basis (9) can be viewed as a decoupled version of (7): it can be readily identified that setting $\alpha = \beta = Z$ and $\mathbf{a} = \mathbf{K}_Z^{-1} \mathbf{m}$ recovers (7). Note that we do not want to define a basis in terms of $\mathbf{K}_\alpha^{-1}$ as that incurs the cubic complexity that we intend to avoid. This basis gives a posterior process with

$$m(x) = \mathbf{k}_{x,\alpha} \mathbf{a}, \qquad s(x, x') = k_{x,x'} - \mathbf{k}_{x,\beta} \mathbf{K}_\beta^{-1} (\mathbf{S} - \mathbf{K}_\beta) \mathbf{K}_\beta^{-1} \mathbf{k}_{\beta,x'}. \tag{10}$$

The alternate choice (9) addresses the difficulty in optimizing the covariance operator, but it still suffers from one serious drawback: while using more inducing points, (9) is not necessarily more expressive than the standard basis (7), for example, when $\alpha$ is selected badly. To eliminate the worst-case setup, we can explicitly consider $\beta$ to be part of $\alpha$ and use

$$\mu = \Psi_\gamma \mathbf{a}_\gamma + \Psi_\beta \mathbf{K}_\beta^{-1} \mathbf{m}_\beta, \qquad \Sigma = (I - \Psi_\beta \mathbf{K}_\beta^{-1} \Psi_\beta^\top) + \Psi_\beta \mathbf{K}_\beta^{-1} \mathbf{S} \mathbf{K}_\beta^{-1} \Psi_\beta^\top. \tag{11}$$

where $\gamma = \alpha \setminus \beta$. This is exactly the *hybrid basis* suggested in the appendix of Cheng and Boots [5], which is strictly more expressive than (7) and yet has the complexity as (8). Also the explicit inclusion of $\beta$ inside $\alpha$ is pivotal to defining proper finite-dimensional natural parameters, which we will later discuss. This basis gives a posterior process with the same covariance as (10), and mean $m(x) = \mathbf{k}_{x,\gamma} \mathbf{a}_\gamma + \mathbf{k}_{x,\beta} \mathbf{K}_\beta^{-1} \mathbf{m}_\beta$.

## 3.2 Orthogonally Decoupled Representation

But is (11) the best possible decoupled basis? Upon closer inspection, we find that there is redundancy in the parameterization of this basis: as $\Psi_\gamma$ is not orthogonal to $\Psi_\beta$ in general, optimizing $\mathbf{a}_\gamma$ and $\mathbf{m}_\beta$ jointly would create coupling and make the optimization landscape more ill-conditioned.

To address this issue, under the partition that $\alpha = \{\beta, \gamma\}$, we propose a new decoupled basis as

$$\mu = (I - \Psi_\beta \mathbf{K}_\beta^{-1} \Psi_\beta^\top) \Psi_\gamma \mathbf{a}_\gamma + \Psi_\beta \mathbf{a}_\beta, \qquad \Sigma = (I - \Psi_\beta \mathbf{K}_\beta^{-1} \Psi_\beta^\top) + \Psi_\beta \mathbf{K}_\beta^{-1} \mathbf{S} \mathbf{K}_\beta^{-1} \Psi_\beta^\top, \tag{12}$$

where $\mathbf{a}_\gamma \in \mathbb{R}^{|\gamma|}, \mathbf{a}_\beta \in \mathbb{R}^{|\beta|}$ and $\mathbf{S} = \mathbf{L}\mathbf{L}^\top$ is parametrized by its Cholesky factor. We call $(\mathbf{a}_\gamma, \mathbf{a}_\beta, \mathbf{S})$ the *model parameters* and refer to (12) as the *orthogonally* decoupled basis, because $(I - \Psi_\beta \mathbf{K}_\beta^{-1} \Psi_\beta^\top)$ is orthogonal to $\Psi_\beta$ (i.e. $(I - \Psi_\beta \mathbf{K}_\beta^{-1} \Psi_\beta^\top)^\top \Psi_\beta = 0$). By substituting $Z = \beta$ and $\mathbf{a}_\beta = \mathbf{K}_Z^{-1} \mathbf{m}$, we can compare (12) to (7): (12) has an additional part parameterized by $\mathbf{a}_\gamma$ to model the mean function residues that *cannot* be captured by using the inducing points $\beta$ alone. In prediction, our basis has time complexity in $O(|\gamma| + |\beta|^3)$ because $\mathbf{K}_\beta^{-1} \mathbf{K}_{\beta,\gamma} \mathbf{a}_\gamma$ can be precomputed. The orthogonally decoupled basis results in a posterior process with

$$m(x) = (\mathbf{k}_{x,\gamma} - \mathbf{k}_{x,\beta} \mathbf{K}_{\beta,\gamma}) \mathbf{a}_\gamma + \mathbf{k}_{x,\beta} \mathbf{a}_\beta, \quad s(x, x') = k_{x,x'} - \mathbf{k}_{x,\beta} \mathbf{K}_\beta^{-1} (\mathbf{S} - \mathbf{K}_\beta^{-1}) \mathbf{K}_\beta^{-1} \mathbf{k}_{\beta,x'}.$$

This decoupled basis can also be derived from the perspective of Titsias [32] by conditioning the prior on a finite set of inducing points*. Details of this construction are in Appendix D.

Compared with the original decoupled basis in (8), our choice in (12) has attractive properties:

1. The explicit inclusion of $\beta$ as a subset of $\alpha$ leads to the existence of natural parameters.
2. If the likelihood is strictly log-concave (e.g. Gaussian and Bernoulli likelihoods), then the variational inference problem in (4) is strictly convex in $(\mathbf{a}_\gamma, \mathbf{a}_\beta, \mathbf{L})$ (see Appendix B).

Our setup in (12) introduces a projection operator before $\Psi_\gamma \mathbf{a}_\gamma$ in the basis (11) and therefore it can be viewed as the *unique* hybrid parametrization, which confines the function modeled by $\gamma$ to be orthogonal to the span the $\beta$ basis. Consequently, there is no correlation between optimizing $\mathbf{a}_\gamma$ and $\mathbf{a}_\beta$, making the problem more well-conditioned.

## 3.3 Natural Parameters and Expectation Parameters

To identify the natural parameter of GPs structured as (12), we revisit the definition of natural parameters in exponential families. A distribution $p(x)$ belongs to an exponential family if we can write $p(x) = h(x)\exp(t(x)^\top \eta - A(\eta))$, where $t(x)$ is the sufficient statistics, $\eta$ is the natural parameter, $A$ is the log-partition function, and $h(x)$ is the carrier measure.

Based on this definition, we can see that the choice of natural parameters is not unique. Suppose $\eta = H\tilde{\eta} + b$ for some constant matrix $H$ and vector $b$. Then $\tilde{\eta}$ is also an admissible natural parameter, because we can write $p(x) = \tilde{h}(x)\exp(\tilde{t}(x)^\top \tilde{\eta} - \tilde{A}(\tilde{\eta}))$, where $\tilde{t}(x) = H^\top t(x)$, $\tilde{h}(x) = h(x)\exp(t(x)^\top b)$, and $\tilde{A}(\tilde{\eta}) = A(H\tilde{\eta} + b)$. In other words, the natural parameter is only unique up to affine transformations. If the natural parameter is transformed, the corresponding expectation parameter $\theta = \mathbb{E}_p[t(x)]$ also transforms accordingly to $\tilde{\theta} = H^\top \theta$. It can be shown that the Legendre primal-dual relationship between $\eta$ and $\theta$ is also preserved: $\tilde{A}$ is also convex, and it satisfies $\tilde{\theta} = \nabla \tilde{A}(\tilde{\eta})$ and $\tilde{\eta} = \nabla \tilde{A}^*(\tilde{\theta})$, where $*$ denotes the Legendre dual function (see Appendix C).

We use this trick to identify the natural and expectation parameters of (12).[*] The relationships between natural, expectation, and model parameters are summarized in Figure 1.

**Natural Parameters** Recall that for Gaussian distributions the natural parameters are conventionally defined as $(\Sigma^{-1}\mu, \frac{1}{2}\Sigma^{-1})$. Therefore, to find the natural parameters of (12), it suffices to show that $(\Sigma^{-1}\mu, \frac{1}{2}\Sigma^{-1})$ of (12) can be written as an affine transformation of some finite-dimensional parameters. The matrix inversion lemma and the orthogonality of $(I - \Psi_\beta \mathbf{K}_\beta^{-1} \Psi_\beta^\top)$ and $\Psi_\beta$ yield

$$\Sigma^{-1}\mu = (I - \Psi_\beta \mathbf{K}_\beta^{-1}\Psi_\beta^\top)\Psi_\gamma \mathbf{j}_\gamma + \Psi_\beta \mathbf{j}_\beta, \qquad \tfrac{1}{2}\Sigma^{-1} = \tfrac{1}{2}(I - \Psi_\beta \mathbf{K}_\beta^{-1}\Psi_\beta^\top) + \Psi_\beta \mathbf{\Theta}\Psi_\beta^\top,$$
$$\text{where} \qquad \mathbf{j}_\gamma = \mathbf{a}_\gamma, \qquad \mathbf{j}_\beta = \mathbf{S}^{-1}\mathbf{K}_\beta \mathbf{a}_\beta, \qquad \mathbf{\Theta} = \tfrac{1}{2}\mathbf{S}^{-1}. \tag{13}$$

Therefore, we call $(\mathbf{j}_\gamma, \mathbf{j}_\beta, \mathbf{\Theta})$ the natural parameters of (12). This choice is unique in the sense that $\Sigma^{-1}\mu$ is orthogonally parametrized.[†] The explicit inclusion of $\beta$ as part of $\alpha$ is important; otherwise there will be a constraint on $\mathbf{j}_\alpha$ and $\mathbf{j}_\beta$ because $\mu$ can only be parametrized by the $\alpha$-basis (see Appendix C).

**Expectation Parameters** Once the new natural parameters are selected, we can also derive the corresponding expectation parameters. Recall for the natural parameters $(\Sigma^{-1}\mu, \frac{1}{2}\Sigma^{-1})$, the associated expectation parameters are $(\mu, -(\Sigma + \mu\mu^\top))$. Using the relationship between transformed natural and expectation parameters, we find the expected parameters of (12) using the adjoint operators: $[(I - \Psi_\beta \mathbf{K}_\beta^{-1}\Psi_\beta^\top)\Psi_\gamma, \Psi_\beta]^\top \mu = [\mathbf{m}_{\gamma\perp\beta}, \mathbf{m}_\beta]^\top$ and $-\Psi_\beta^\top(\Sigma + \mu\mu^\top)\Psi_\beta = \mathbf{\Lambda}$, where we have

$$\mathbf{m}_{\gamma\perp\beta} = (\mathbf{K}_\gamma - \mathbf{K}_{\gamma,\beta}\mathbf{K}_\beta^{-1}\mathbf{K}_{\beta,\gamma})\mathbf{j}_\gamma, \qquad \mathbf{m}_\beta = \mathbf{S}\mathbf{j}_\beta, \qquad \mathbf{\Lambda} = -\mathbf{S} - \mathbf{m}_\beta \mathbf{m}_\beta^\top. \tag{14}$$

Note the equations for $\beta$ in (13) and (14) have exactly the same relationship between the natural and expectation parameters in the standard Gaussian case, i.e. $(\Sigma^{-1}\mu, \frac{1}{2}\Sigma^{-1}) \leftrightarrow (\mu, -(\Sigma + \mu\mu^\top))$.

## 3.4 Natural Gradient Descent

Natural gradient descent updates parameters according to the information geometry induced by the KL divergence [2]. It is invariant to reparametrization and can normalize the problem to be well conditioned [20]. Let $F(\eta) = \nabla^2 \mathrm{KL}(q||p_\eta)|_{q=p_\eta}$ be the Fisher information matrix, where $p_\eta$ denotes a distribution with natural parameter $\eta$. Natural gradient descent for natural parameters performs the update $\eta \leftarrow \eta - \tau F(\eta)^{-1}\nabla_\eta \mathcal{L}$, where $\tau > 0$ is the step size. Because directly computing the inverse $F(\eta)^{-1}$ is computationally expensive, we use the duality between natural and expectation parameters in exponential families and adopt the equivalent update $\eta \leftarrow \eta - \tau \nabla_\theta \mathcal{L}$ [15, 29].

---

[*]While GPs do not admit a density function, the property of transforming natural parameters described above still applies. An alternate proof can be derived using KL divergence.

[†]The hybrid parameterization (11) in [5, Appendix], which also considers $\beta$ explicitly in $\mu$, admits natural parameters as well. However, their relationship and the natural gradient update rule turn out to be more convoluted; we provide a thorough discussion in Appendix C.

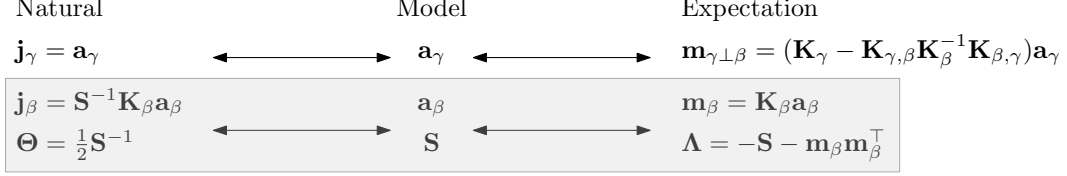

| Natural | Model | Expectation |
|---------|-------|-------------|
| $\mathbf{j}_\gamma = \mathbf{a}_\gamma$ | $\mathbf{a}_\gamma$ | $\mathbf{m}_{\gamma\perp\beta} = (\mathbf{K}_\gamma - \mathbf{K}_{\gamma,\beta}\mathbf{K}_\beta^{-1}\mathbf{K}_{\beta,\gamma})\mathbf{a}_\gamma$ |
| $\mathbf{j}_\beta = \mathbf{S}^{-1}\mathbf{K}_\beta\mathbf{a}_\beta$ | $\mathbf{a}_\beta$ | $\mathbf{m}_\beta = \mathbf{K}_\beta\mathbf{a}_\beta$ |
| $\boldsymbol{\Theta} = \frac{1}{2}\mathbf{S}^{-1}$ | $\mathbf{S}$ | $\boldsymbol{\Lambda} = -\mathbf{S} - \mathbf{m}_\beta\mathbf{m}_\beta^\top$ |

Figure 1: The relationship between the three parameterizations of the orthogonally decoupled basis. The box highlights the parameters in common with the standard coupled basis, which are decoupled from the additional $\mathbf{a}_\gamma$ parameter. This is a unique property of our orthogonal basis

**Exact Update Rules**  For our basis in (12), the natural gradient descent step can be written as

$$\mathbf{j} \leftarrow \mathbf{j} - \tau\nabla_{\mathbf{m}}\mathcal{L}, \qquad \boldsymbol{\Theta} \leftarrow \boldsymbol{\Theta} - \tau\nabla_{\boldsymbol{\Lambda}}\mathcal{L}, \tag{15}$$

where we recall $\mathcal{L}$ is the negative variational lower bound in (4), $\mathbf{j} = [\mathbf{j}_\gamma, \mathbf{j}_\beta]$ in (13), and $\mathbf{m} = [\mathbf{m}_{\gamma\perp\beta}, \mathbf{m}_\beta]$ in (14). As $\mathcal{L}$ is defined in terms of $(\mathbf{a}_\gamma, \mathbf{a}_\beta, \mathbf{S})$, to compute these derivatives we use chain rule (provided by the relationship in Figure 1) and obtain

$$\mathbf{j}_\gamma \leftarrow \mathbf{j}_\gamma - \tau(\mathbf{K}_\gamma - \mathbf{K}_{\gamma,\beta}\mathbf{K}_\beta^{-1}\mathbf{K}_{\gamma,\beta})^{-1}\nabla_{\mathbf{a}_\gamma}\mathcal{L}, \tag{16a}$$

$$\mathbf{j}_\beta \leftarrow \mathbf{j}_\beta - \tau(\mathbf{K}_\beta^{-1}\nabla_{\mathbf{a}_\beta}\mathcal{L} - 2\nabla_{\mathbf{S}}\mathcal{L}\mathbf{m}_\beta), \tag{16b}$$

$$\boldsymbol{\Theta} \leftarrow \boldsymbol{\Theta} + \tau\nabla_{\mathbf{S}}\mathcal{L}. \tag{16c}$$

Due to the orthogonal choice of natural parameter definition, the update for the $\gamma$ and the $\beta$ parts are independent. Furthermore, one can show that the update for $\mathbf{j}_\beta$ and $\boldsymbol{\Theta}$ is exactly the same as the natural gradient descent rule for the standard coupled basis [12], and that the update for the residue part $\mathbf{j}_\gamma$ is equivalent to functional gradient descent [17] in the subspace orthogonal to the span of $\Psi_\beta$.

**Approximate Update Rule**  We described the natural gradient descent update for the orthogonally decoupled GPs in (12). However, in the regime where $|\gamma| \gg |\beta|$, computing (16a) becomes infeasible. Here we propose an approximation of (16a) by approximating $\mathbf{K}_\gamma$ with a diagonal-plus-low-rank structure. Because the inducing points $\beta$ are selected to globally approximate the function landscape, one sensible choice is to approximate $\mathbf{K}_\gamma$ with a Nyström approximation based on $\beta$ and a diagonal correction term: $\mathbf{K}_\gamma \approx \mathbf{D}_{\gamma|\beta} + \mathbf{K}_{\gamma|\beta}$, where $\mathbf{D}_{\gamma|\beta} = \mathrm{diag}(\mathbf{K}_\gamma - \mathbf{K}_{\gamma|\beta})$, $\mathbf{K}_{\gamma|\beta} = \mathbf{K}_{\gamma,\beta}\mathbf{K}_\beta^{-1}\mathbf{K}_{\beta,\gamma}$, and $\mathrm{diag}$ denotes extracting the diagonal part of a matrix. FITC [30] uses a similar idea to approximate the prior distribution [25], whereas here it is used to derive an approximate update rule without changing the problem. This leads to a simple update rule

$$\mathbf{j}_\gamma \leftarrow \mathbf{j}_\gamma - \tau(\mathbf{D}_{\gamma|\beta} + \epsilon\mathbf{I})^{-1}\nabla_{\mathbf{a}_\gamma}\mathcal{L}, \tag{17}$$

where a jitter $\epsilon > 0$ is added to ensure stability. This update rule uses a diagonal scaling $(\mathbf{D}_{\gamma|\beta} + \epsilon\mathbf{I})^{-1}$, which is independent of $\mathbf{j}_\beta$ and $\boldsymbol{\Theta}$. Therefore, while one could directly use the update (17), in implementation, we propose to replace (17) with an adaptive coordinate-wise gradient descent algorithm (e.g. ADAM [16]) to update the $\gamma$-part. Due to the orthogonal structure, the overall computational complexity is in $O(|\gamma||\beta| + |\beta|^3)$. While this is more than the $O(|\gamma| + |\beta|^3)$ complexity of the original decoupled approach [5]; the experimental results suggest the additional computation is worth the large performance improvement.

## 4  Results

We empirically assess the performance of our algorithm in multiple regression and classification tasks. We show that *a*) given fixed wall-clock time, the proposed orthogonally decoupled basis outperforms existing approaches; *b*) given the same number of inducing points for the covariance, our method almost always improves on the coupled approach (which is in contrast to the previous decoupled bases); *c*) using natural gradients can dramatically improve performance, especially in regression.

We compare updating our orthogonally decoupled basis with adaptive gradient descent using the Adam optimizer [16] (ORTH), and using the approximate natural gradient descent rule described in Section 3.4 (ORTHNAT). As baselines, we consider the original decoupled approach of Cheng and Boots

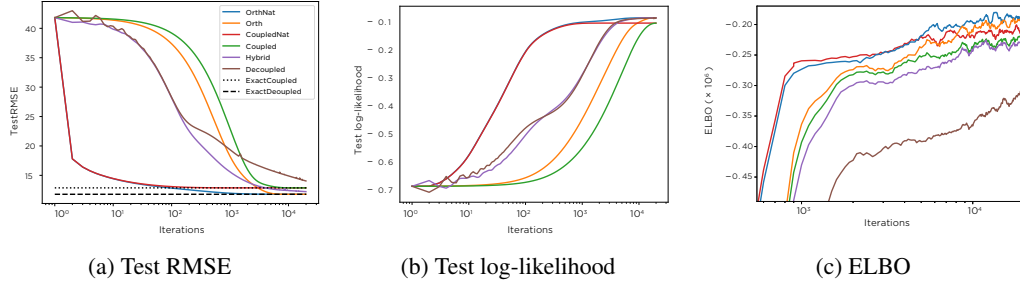

|                     |                     |                     |
|:-------------------:|:-------------------:|:-------------------:|
| (a) Test RMSE       | (b) Test log-likelihood | (c) ELBO        |

Figure 2: Training curves for our models in three different settings. Panel (c) has $|\gamma| = 3500, |\beta| = 1500$ for the decoupled bases and $|\beta| = 2000$ for the coupled bases. Panels (a) and (b) have $|\gamma| = |\beta| = 500$ and fixed hyperparameters and full batches to highlight the convergence properties of the approaches. Panels (a) and (c) use a Gaussian likelihood. Panel (b) uses a Bernoulli likelihood.

[5] (DECOUPLED) and the hybrid approach suggested in their Appendix (HYBRID). We compare also to the standard coupled basis with and without natural gradients (COUPLEDNAT and COUPLED, respectively). We make generic choices for hyperparameters, inducing point initializations, and data processing, which are detailed in Appendix F. Our code [*] and datasets [†] are publicly available.

**Illustration**   Figures 2a and 2b show a simplified setting to illustrate the difference between the methods. In this example, we fixed the inducing inputs and hyperparameters, and optimized the rest of the variational parameters. All the decoupled methods then have the same global optimum, so we can easily assess their convergence property. With a Gaussian likelihood we also computed the optimal solution analytically as an optimal baseline, although this requires inverting an $|\alpha|$-sized matrix, and therefore is not useful as a practical method. We include the expressions of the optimal solution in Appendix E. We set $|\gamma| = |\beta| = 500$ for all bases and conducted experiments on `3droad` dataset ($N = 434874$, $D = 3$) for regression with a Gaussian likelihood and `ringnorm` data ($N = 7400$, $D = 21$) for classification with a Bernoulli likelihood. Overall, the natural gradient methods are much faster to converge than their ordinary gradient counterparts. DECOUPLED fails to converge to the optimum after 20K iterations, even in the Gaussian case. We emphasize that, unlike our proposed approaches, DECOUPLED leads to a non-convex optimization problem.

**Wall-clock comparison**   To investigate large-scale performance, we used `3droad` with a large number of inducing points. We used a computer with a Tesla K40 GPU and found that, in wall-clock time, the orthogonally decoupled basis with $|\gamma| = 3500$, $|\beta| = 1500$ was equivalent to a coupled model with $|\beta| = 2000$ (about 0.7 seconds per iteration) in our tensorflow [1] implementation. Under this setting, we show the ELBO in Figure 2c and the test log-likelihood and accuracy in Figure 3 of the Appendix G. ORTHNAT performs the best, both in terms of log-likelihood and accuracy. The the highest test log-likelihood of ORTHNAT is $-3.25$, followed by ORTH ($-3.26$). COUPLED ($-3.37$) and COUPLEDNAT ($-3.33$) both outperform HYBRID ($-3.39$) and DECOUPLED ($-3.66$).

**Regression benchmarks**   We applied our models on 12 regression datasets ranging from 15K to 2M points. To enable feasible computation on multiple datasets, we downscale (but keep the same ratio) the number of inducing points to $|\gamma| = 700, |\beta| = 300$ for the decoupled models, and $|\beta| = 400$ for the coupled mode. We compare also to the coupled model with $|\beta| = 300$ to establish whether extra computation always improves the performance of the decoupled basis. The test mean absolute error (MAE) results are summarized in Table 1, and the full results for both test log-likelihood and MAE are given in Appendix G. ORTHNAT overall is the most competitive basis. And, by all measures, the orthogonal bases outperform their coupled counterparts with the same $\beta$, except for HYBRID and DECOUPLED.

**Classification benchmarks**   We compare our method with state-of-the-art fully connected neural networks with Selu activations [18]. We adopted the experimental setup from [18], using the largest 19 datasets (4898 to 130000 data points). For the binary datasets we used the Bernoulli likelihood, and

---

[*] https://github.com/hughsalimbeni/orth_decoupled_var_gps
[†] https://github.com/hughsalimbeni/bayesian_benchmarks

Table 1: Regression summary for normalized test MAE on 12 regression datasets, with standard errors for the average ranks. The coupled bases had $|\beta| = 400$ ($|\beta| = 300$ for the † bases), and the decoupled all had $\gamma = 700$, $\beta = 300$. See Appendix G for the full results.

|  | COUPLED† | COUPLEDNAT† | COUPLED | COUPLEDNAT | ORTH | ORTHNAT | HYBRID | DECOUPLED |
|---|---|---|---|---|---|---|---|---|
| Mean | 0.298 | 0.295 | 0.291 | 0.290 | 0.284 | **0.282** | 0.298 | 0.361 |
| Median | 0.221 | 0.219 | 0.215 | 0.213 | 0.211 | **0.210** | 0.225 | 0.299 |
| Avg Rank | 6.083(0.19) | 5.00(0.33) | 3.750(0.26) | 2.417(0.31) | 2.500(0.47) | **1.833(0.35)** | 6.417(0.23) | 8(0.00) |

for the multiclass datasets we used the robust-max likelihood [14]. The same basis settings as for the regression benchmarks were used here. ORTH performs the best in terms of median, and ORTHNAT is best ranked. The neural network wins in terms of mean, because it substantially outperforms all the GP models in one particular dataset (`chess-krvk`), which skews the mean performance over the 19 datasets. We see that our orthogonal bases on average improve the coupled bases with equivalent wall-clock time, although for some datasets the coupled bases are superior. Unlike in the regression case, it is not always true that using natural gradients improve performance, although on average they do. This holds for both the coupled and decoupled bases.

Table 2: Classification test accuracy(%) results for our models, showing also the results from [18], with standard errors for the average ranks. See Table 5 in the Appendix for the complete results.

|  | Selu | COUPLED | COUPLEDNAT | ORTH | ORTHNAT | HYBRID | DECOUPLED |
|---|---|---|---|---|---|---|---|
| Mean | **91.6** | 90.4 | 90.2 | 90.6 | 90.3 | 89.9 | 89.0 |
| Median | 93.1 | 94.8 | 93.6 | **95.6** | 93.6 | 93.4 | 92.0 |
| Average rank | 4.16(0.67) | 3.89(0.42) | 3.53(0.45) | 3.68(0.35) | **3.42(0.31)** | 3.89(0.38) | 5.42(0.51) |

Overall, the empirical results demonstrate that the orthogonally decoupled basis is superior to the coupled basis with the same wall-clock time, averaged over datasets. It is important to note that for the *same* $\beta$, adding extra $\gamma$ increases performance for the orthogonally decoupled basis in almost all cases, but not for HYBRID of DECOUPLED. While this does add additional computation, the ratio between the extra computation for additional $\beta$ and that for additional $\gamma$ decreases to zero as $\beta$ increases. That is, eventually the cubic scaling in $|\beta|$ will dominate the linear scaling in $|\gamma|$.

## 5 Conclusion

We present a novel orthogonally decoupled basis for variational inference in GP models. Our basis is constructed by extending the standard coupled basis with an additional component to model the mean residues. Therefore, it extends the standard coupled basis [32, 12] and achieves better performance. We show how the natural parameters of our decoupled basis can be identified and propose an approximate natural gradient update rule, which significantly improves the optimization performance over original decoupled approach [5]. Empirically, our method demonstrates strong performance in multiple regression and classification tasks.

## Footnotes

*We thank an anonymous reviewer for highlighting this connection.

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
