[Supplementary Material]

# Appendix

## A Variational Inference Problem

In this section, we provides details of implementing the variational inference problem

$$\mathcal{L}(q) = -\sum_{n=1}^{N} \mathbb{E}_{q(f(x_n))}[\log p(y_n|f(x_n))] + \text{KL}(q||p) \tag{4}$$

when the variational posterior $q(f) = \mathcal{GP}_{\mathcal{H}}(\mu, \Sigma)$ is parameterized using a decoupled basis

$$\mu = \Psi_\alpha \mathbf{a}, \qquad \Sigma = I + \Psi_\beta \mathbf{A} \Psi_\beta^\top. \tag{5}$$

Without loss of generality, we assume $\mathbf{A} = \mathbf{K}_\beta^{-1}\mathbf{S}\mathbf{K}_\beta^{-1} - \mathbf{K}_\beta^{-1}$. That is, we focus on the following form of parametrization with $\mathbf{S} \succeq 0$,

$$\mu = \Psi_\alpha \mathbf{a}, \qquad \Sigma = I + \Psi_\beta \mathbf{A} \Psi_\beta^\top := (I - \Psi_\beta \mathbf{K}_\beta^{-1} \Psi_\beta^\top) + \Psi_\beta \mathbf{K}_\beta^{-1}\mathbf{S}\mathbf{K}_\beta^{-1}\Psi_\beta^\top. \tag{18}$$

### A.1 KL Divergence

We first show how the KL divergence can be computed using finite-dimensional variables. The proof is similar to strategy in [5, Appendix]

**Proposition A.1.** *For $p = \mathcal{GP}_{\mathcal{H}}(0, I)$ and $q(f) = \mathcal{GP}_{\mathcal{H}}(\mu, \Sigma)$ with*

$$\mu = \Psi_\alpha \mathbf{a}, \qquad \Sigma = \left(I - \Psi_\beta \mathbf{K}_\beta^{-1} \Psi_\beta^\top\right) + \Psi_\beta \mathbf{K}_\beta^{-1}\mathbf{S}\mathbf{K}_\beta^{-1}\Psi_\beta^\top$$

*Then It satisfies*

$$\text{KL}(q||p) = \frac{1}{2}\left(\mathbf{a}^\top \mathbf{K}_\alpha \mathbf{a} + \text{tr}\left(\mathbf{S}\mathbf{K}_\beta^{-1}\right) - \log|\mathbf{S}| + \log|\mathbf{K}_\beta| - |\beta|\right)$$

For the orthogonally decoupled basis (12) in particular, we can write

$$\text{KL}(q||p) = \frac{1}{2}\left(\mathbf{a}_\gamma^\top(\mathbf{K}_\gamma - \mathbf{K}_{\gamma,\beta}\mathbf{K}_\beta^{-1}\mathbf{K}_{\beta,\gamma})\mathbf{a}_\gamma + \mathbf{a}_\beta^\top \mathbf{K}_\beta \mathbf{a}_\beta + \text{tr}\left(\mathbf{S}\mathbf{K}_\beta^{-1}\right) - \log|\mathbf{S}| + \log|\mathbf{K}_\beta| - |\beta|\right).$$

### A.2 Expected Log-Likelihoods

The expected log-likelihood can be computed by first computing the predictive Gaussian distribution $q(f(x)) = \mathcal{N}(f(x)|m(x), s(x))$ for each data point $x$. For example, for the orthogonally decoupled basis (12), this is given as

$$m(x) = (\mathbf{k}_{x,\gamma} - \mathbf{k}_{x,\beta}\mathbf{K}_\beta^{-1}\mathbf{K}_{\beta,\gamma})\mathbf{a}_\gamma + \mathbf{k}_{x,\beta}\mathbf{a}_\beta$$

$$s(x) = (k_x - \mathbf{k}_{x,\beta}\mathbf{K}_\beta^{-1}\mathbf{k}_{\beta,x}) + \mathbf{k}_{x,\beta}\mathbf{K}_\beta^{-1}\mathbf{S}\mathbf{K}_\beta^{-1}\mathbf{k}_{\beta,x}.$$

Given $m(x)$ and $s(x)$, then the expected log-likelihood can be computed exactly (for Gaussian case) or using quadrature approximation.

### A.3 Gradient Computation

Using the above formulas, a differentiable computational graph can be constructed and then the gradient can to $(\mathbf{a}_\gamma, \mathbf{a}_\beta, \mathbf{L})$ can be computed using automatic differentiation. When $\mathbf{a}_\gamma^\top \mathbf{K}_\gamma \mathbf{a}_\gamma$ in the KL-divergence is further approximated by column sampling $\mathbf{K}_\gamma$, an unbiased gradient can be computed in time complexity $O(|\gamma||\beta| + |\beta|^3)$.

## B Convexity of the Variational Inference Problem

Here we show the objective function in (4) is strictly convex in $(\mathbf{a}_\gamma, \mathbf{a}_\beta, \mathbf{L})$ if the likelihood is log-strictly-convex.

## B.1 KL Divergence

We first study the KL divergence term. It is easy to see that it is strongly convex in $(\mathbf{a}_\gamma, \mathbf{a}_\beta)$. When $\mathbf{S} = \mathbf{L}\mathbf{L}^\top$, where $\mathbf{L}$ is lower triangle and with positive diagonal terms, the KL divergence is strongly convex in $\mathbf{L}$ as well. To see this, we notice that

$$-\log|\mathbf{S}| = -\log|\mathbf{L}\mathbf{L}^\top| = -2\sum_{i=1}^{|\beta|}\log|L_{ii}|$$

is strictly convex and

$$\mathrm{tr}\left(\mathbf{K}_\beta^{-1}\mathbf{S}\right) = \mathrm{tr}\left(\mathbf{L}\mathbf{L}^\top\mathbf{K}_\beta^{-1}\right) = \mathrm{vec}(\mathbf{L})^\top(\mathbf{I}\otimes\mathbf{K}_\beta^{-1})\mathrm{vec}(\mathbf{L})$$

is strongly convex, because $\mathbf{K}_\beta^{-1} \succ 0$.

## B.2 Expected Log-likelihood

Here we show the negative expected log-likelihood part is strictly convex. For the negative expected log-likelihood, let $F(\cdot) = -\log(y_n|\cdot)$ and we can write

$$\mathcal{E}_n = \mathbb{E}_{q(f(x_n))}[-\log p(y_n|f(x_n))]$$
$$= \mathbb{E}_{\zeta,\boldsymbol{\xi}}[F(m(x_n) + \zeta + \mathbf{k}_{x_n,\beta}\mathbf{K}_\beta^{-1}\mathbf{L}\boldsymbol{\xi})]$$

in which $\zeta \sim \mathcal{N}(\zeta|0, k_{x_n} - \mathbf{k}_{x_n,\beta}\mathbf{K}_\beta^{-1}\mathbf{k}_{\beta,x_n})$ and $\boldsymbol{\xi} \sim \mathcal{N}(\boldsymbol{\xi}|0,\mathbf{I})$.

Then we give a lemma below.

**Lemma B.1.** *Suppose $f$ is $\theta$-strictly convex. Then $f(Ax)$ is also $\theta$-strictly convex.*

*Proof.* Let $u = Ax$ and $v = Ay$. Let $g(x) = f(Ax)$.

$$f(v) - f(u) \geq \langle \nabla f(u), v - u\rangle + \frac{\theta}{2}(\langle \nabla f(u), v - u\rangle)^2$$
$$= \langle \nabla f(u), A(y - x)\rangle + \frac{\theta}{2}(\langle \nabla f(u), A(y - x)\rangle)^2$$
$$= \langle A^\top \nabla f(u), y - x\rangle + \frac{\theta}{2}(\langle A^\top \nabla f(u), y - x\rangle)^2$$
$$= \langle \nabla g(x), y - x\rangle + \frac{\theta}{2}(\langle \nabla g(x), y - x\rangle)^2 \qquad \blacksquare$$

Because $F$ is strictly convex when likelihood is log-strictly-concave and $m(x_n)$ is linearly parametrized, the desired strict convexity follows.

# C Uniqueness of Parametrization and Natural Parameters

Here we provide some additional details regarding natural parameters and natural gradient descent.

## C.1 Necessity of Including $\beta$ as Subset of $\alpha$

We show that the partition condition in Section 3.2 is necessary to derive proper natural parameters. Suppose the contrary case where $\alpha$ is a general set of inducing points. Using a similar derivation as Section 3.3, we show that

$$\frac{1}{2}\Sigma^{-1} = \frac{1}{2}\left(I - \Psi_\beta\mathbf{K}_\beta^{-1}\Psi_\beta^\top\right) + \frac{1}{2}\Psi_\beta\mathbf{S}^{-1}\Psi_\beta^\top$$
$$\Sigma^{-1}\mu = \left(\left(I - \Psi_\beta\mathbf{K}_\beta^{-1}\Psi_\beta^\top\right) + \Psi_\beta\mathbf{S}^{-1}\Psi_\beta^\top\right)\Psi_\alpha\mathbf{a}$$
$$= (I - \Psi_\beta\mathbf{K}_\beta^{-1}\Psi_\beta^\top)\Psi_\alpha\tilde{\mathbf{j}}_\alpha + \Psi_\beta\tilde{\mathbf{j}}_\beta$$

where $\tilde{\mathbf{j}}_\alpha = \mathbf{a}_\alpha$ and $\tilde{\mathbf{j}}_\beta = \mathbf{S}^{-1}\mathbf{K}_\beta\mathbf{a}_\alpha$. Therefore, we might consider choosing $(\mathbf{j}_\alpha, \mathbf{j}_\beta, \frac{1}{2}\mathbf{S}^{-1})$ as a candidate for natural parameters. However the above choice of parametrization is actually coupled due to the condition that $\mathbf{j}_\alpha$ and $\mathbf{j}_\beta$ have to satisfy, i.e.

$$\tilde{\mathbf{j}}_\beta = \mathbf{S}^{-1}\mathbf{K}_\beta\tilde{\mathbf{j}}_\alpha$$

Thus, they cannot satisfy the requirement of being natural parameters. This is mainly because $\mu$ is given in only $\alpha$ basis, whereas $\Sigma^{-1}\mu$ is given in both $\alpha$ and $\beta$ bases.

## C.2 Alternate Choices of Natural Parameters

As discussed previously in Section 3.3, the choice of natural parameters is only unique up to affine transformation. While in this paper we propose to use the unique orthogonal version, other choices of parametrization are possible. For instance, here we consider the hybrid parametrization in [5, appendix] and give an overview on finding its natural parameters.

The hybrid parametrization use the following decoupled basis:

$$\mu = \Psi_\gamma\mathbf{a}_\gamma + \Psi_\beta\mathbf{a}_\beta \qquad \Sigma = (I - \Psi_\beta\mathbf{K}_\beta^{-1}\Psi_\beta^\top) + \Psi_\beta\mathbf{K}_\beta^{-1}\mathbf{S}\mathbf{K}_\beta^{-1}\Psi_\beta^\top$$

To facilitate a clear comparison, here we remove the $\mathbf{K}_\beta^{-1}$ in the original form suggested by Cheng and Boots [5], which uses $\mu = \Psi_\gamma\mathbf{a}_\gamma + \Psi_\beta\mathbf{K}_\beta^{-1}\mathbf{a}_\beta$. Note in the experiments, their original form was used.

As the covariance part above is the same form as our orthogonally decoupled basis in (12), here we only consider the mean part. Following a similar derivation, we can write

$$\begin{aligned}
\Sigma^{-1}\mu &= \left(\left(I - \Psi_\beta\mathbf{K}_\beta^{-1}\Psi_\beta^\top\right) + \Psi_\beta\mathbf{S}^{-1}\Psi_\beta^\top\right)(\Psi_\gamma\mathbf{a}_\gamma + \Psi_\beta\mathbf{a}_\beta) \\
&= \Psi_\gamma\mathbf{a}_\gamma + \Psi_\beta(\mathbf{S}^{-1} - \mathbf{K}_\beta^{-1})\mathbf{K}_{\beta,\gamma}\mathbf{a}_\gamma + \Psi_\beta\mathbf{S}^{-1}\mathbf{K}_\beta\mathbf{a}_\beta \\
&= (\Psi_\gamma - \Psi_\beta\mathbf{K}_\beta^{-1}\mathbf{K}_{\beta,\gamma})\mathbf{a}_\gamma + \Psi_\beta\mathbf{S}^{-1}(\mathbf{K}_\beta\mathbf{a}_\beta + \mathbf{K}_{\beta,\gamma}\mathbf{a}_\gamma) \\
&= (I - \Psi_\beta\mathbf{K}_\beta^{-1}\Psi_\beta^\top)\Psi_\gamma\mathbf{j}_\gamma + \Psi_\beta\mathbf{j}_\beta
\end{aligned}$$

That is, we can choose the natural parameters as

$$\mathbf{j}_\gamma = \mathbf{a}_\gamma, \qquad \mathbf{j}_\beta = \mathbf{S}^{-1}(\mathbf{K}_\beta\mathbf{a}_\beta + \mathbf{K}_{\beta,\gamma}\mathbf{a}_\gamma), \qquad \mathbf{\Theta} = \frac{1}{2}\mathbf{S}^{-1} \tag{19}$$

This set of natural parameters, unlike the one in the previous section, is proper, because $\beta$ included as a subset of $\alpha$.

Comparing (19) with (13), we can see that there is a coupling between $\mathbf{a}_\gamma$ and $\mathbf{j}_\beta$ in (19). This would lead to a more complicated update rule in computing the natural gradient. This coupling phenomenon also applies to other choice of parametrizations, excerpt for our orthogonally decoupled basis.

## C.3 Invariance of Natural Gradient Descent

As discussed above, the choice of natural parameters for the mean part is not unique, but here we show they all lead to the same natural gradient descent update. Therefore, our orthogonal choice (12), among all possible equivalent parameterizations, has the cleanest update rule.

This equivalence between different parameterizations can be easily seen from that the KL divergence between Gaussians are quadratic in $\Sigma^{-1}\mu$. Therefore, the natural gradient of $\Sigma^{-1}\mu$ has the form as the proximal update below

$$\arg\min_x \langle\nabla_x f, x\rangle + \frac{1}{2}(x - y)^\top Q(x - y) = y - Q^{-1}g$$

for some function $f$, vector $y$ and positive-definite matrix $Q$.

To see the invariance of invertible linear transformations, suppose we reparametrize $x, y$ above as $x = Au + b$ and $y = Av + b$, for some invertible $A$ and $b$. Then the update becomes

$$\arg\min_u \langle \nabla_z f, u \rangle + \frac{1}{2}(u - v)^\top A^\top Q A (u - v)$$

$$\arg\min_u \langle A^\top \nabla_x f, u \rangle + \frac{1}{2}(u - v)^\top A^\top Q A (u - v)$$

$$= v - A^{-1} Q^{-1} \nabla_x f$$

which represents the same update step in $x$ because

$$A(v - A^{-1} Q^{-1} \nabla_x f) + b = y - Q^{-1} \nabla_x f.$$

## C.4 Transformation of Natural Parameters and Expectation Parameters

Here we provide a more rigorous proof of identifying natural and expectation parameters of decoupled bases, as the density function $p(f)$, which is used to illustrate the idea in Section 3.3, is not defined for GPs. Here we show the transformation of natural parameters and expectation parameters based on KL divergence. We start from $d$-dimensional exponential families and then show that the formulation extends to arbitrary $d$.

Consider a $d$-dimensional exponential family. Its KL divergence of an exponential family can be written as

$$\text{KL}(q\|p) = A(\eta_p) + A^*(\theta_q) - \langle \eta_p, \theta_q \rangle \tag{20}$$

where $A$ is the log-partition function, $A^*$ is its Legendre dual of $A$, $\theta$ is the expectation parameter, and $\eta$ is the natural parameter. It holds the duality property that $\theta_p = \nabla A(\eta_p)$ and $\eta_p = \nabla A^*(\theta_p)$.

As (20) is expressed in terms of inner product, it holds for arbitrary $d$ and it is defined finitely for GPs with decoupled basis [5]. Therefore, here we show that when we parametrize problem by $\eta_p = H\tilde{\eta}_p + b$, $\tilde{\eta}_p$ is also a candidate natural parameter satisfying (20) for some transformed expectation parameter $\tilde{\theta}_q$. It can be shown as below

$$
\begin{aligned}
\text{KL}(q\|p) &= A(\eta_p) + A^*(\theta_q) - \langle \eta_p, \theta_q \rangle \\
&= A(H\tilde{\eta}_p + b) + A^*(\theta_q) - \langle H\tilde{\eta}_p + b, \theta_q \rangle \\
&= A(H\tilde{\eta}_p + b) + A^*(\theta_q) - \langle \tilde{\eta}_p, H^\top \theta_q \rangle - \langle b, \theta_q \rangle \\
&= A(H\tilde{\eta}_p + b) + \left( A^*(H^{-\top}\tilde{\theta}_q) - \left\langle b, H^{-\top}\tilde{\theta}_q \right\rangle \right) - \left\langle \tilde{\eta}_p, \tilde{\theta}_q \right\rangle \\
&=: \tilde{A}(\tilde{\eta}_p) + \tilde{A}^*(\tilde{\theta}_q) - \left\langle \tilde{\eta}_p, \tilde{\theta}_h \right\rangle
\end{aligned}
$$

where we define

$$\tilde{\theta}_q = H^\top \theta_q$$

$$\tilde{A}(\tilde{\eta}_p) = A(H\tilde{\eta}_p + b)$$

$$\tilde{A}^*(\tilde{\theta}_q) = A^*(H^{-\top}\tilde{\theta}_q) - \left\langle b, H^{-\top}\tilde{\theta}_q \right\rangle$$

It can be verified that $\tilde{A}^*$ is indeed the Legendre dual of $\tilde{A}$.

$$
\begin{aligned}
\max_x \langle w, x \rangle - \tilde{A}(x) &= \max_x \langle w, x \rangle - A(Hx + b) \\
&= \max_z \left\langle w, H^{-1}(z - b) \right\rangle - A(z) \\
&= -\left\langle H^{-\top}w, b \right\rangle + \max_z \left\langle H^{-\top}w, z \right\rangle - A(z) \\
&= -\left\langle H^{-\top}w, b \right\rangle + A^*(H^{-\top}w) = \tilde{A}^*(w)
\end{aligned}
$$

Note the inversion requirement on $H$ can be removed by replacing $-\top$ with pseudo-inverse, because $\tilde{\theta}_q$ lies in the range of $H^\top$. Thus, if $\eta = H\tilde{\eta} + b$ and $\theta$ are one choice of natural-expectation parameter pair, then $\tilde{\eta}$ and $\tilde{\theta} = H^T \eta$ is another natural-expectation parameter pair.

# D    Primal Representation of Orthogonally Decoupled GPs

In this section, we demonstrate that the orthogonally decoupled GPs have an equivalent construction from the primal viewpoint adopted by variational inference framework of Titsias [32]. The key idea is to use two sets of inducing points. We use them to form a posterior process by conditioning the prior like the usual way, but in the meantime imposing a particular restriction on the variational distribution at the inducing points.

## D.1    The Variational Posterior Process Proposed by Titsias [32]

The approach of Titsias [32] begins with expressing the prior process in terms of the following factorization*:

$$p(f) = p(f|\mathbf{f}_\beta)p(\mathbf{f}_\beta),$$

where $\mathbf{f}_\beta$ are function values at locations $\beta$, often referred to as "inducing points." For simplicity we assume zero prior mean, so the prior at the inducing points is $p(\mathbf{f}_\beta) = \mathcal{N}(\mathbf{f}_\beta|\mathbf{0}, \mathbf{K}_\beta)$ and the prior conditional process $p(f|\mathbf{f}_\beta)$ is a GP which we denote as $\mathcal{GP}(m_{\mathbf{f}_\beta}, s_{\mathbf{f}_\beta})$ with

$$m_{\mathbf{f}_\beta}(x) = \mathbf{k}_{x,\beta}^\top \mathbf{K}_\beta^{-1} \mathbf{f}_\beta$$
$$s_{\mathbf{f}_\beta}(x, x') = k(x, x') - \mathbf{k}_{x,\beta}^\top \mathbf{K}_\beta^{-1} \mathbf{k}_{\beta,x'}$$

The key idea of Titsias [32], which is later developed by [12, 22], is to define the variational posterior process as

$$q(f) = p(f|\mathbf{f}_\beta)q(\mathbf{f}_\beta), \tag{21}$$

where $q(\mathbf{f}_\beta) = \mathcal{N}(\mathbf{f}_\beta|\mathbf{m}_\beta, \mathbf{S}_\beta)$ for some variational parameters $\mathbf{m}_\beta$ and $\mathbf{S}_\beta$. Since the conditional process is linear in $\mathbf{f}_\beta$ and $q(\mathbf{f}_\beta)$ is Gaussian, we can use standard properties for Gaussians (i.e., $\int_x \mathcal{N}(y|a + Lx, A)\mathcal{N}(x|b, B)dx \propto \mathcal{N}(y|a + Lb, A + LBL^\top))$ to derive the mean and covariance functions of the variational posterior process $q(f)$ in (21):

$$m(x) = \mathbf{k}_{x,\beta} \mathbf{K}_\beta^{-1} \mathbf{m}_\beta \tag{22}$$
$$s(x, x') = k(x, x') + \mathbf{k}_{x,\beta} \mathbf{K}_\beta^{-1} (\mathbf{S}_\beta - \mathbf{K}_\beta) \mathbf{K}_\beta^{-1} \mathbf{k}_{\beta,x'} \tag{23}$$

## D.2    The Equivalent Posterior Process of the Orthogonally Decoupled Basis

To derive our orthogonally decoupled approach, we introduce further a set of disjoint inducing points denoted as $\gamma$. Let $\mathbf{f}_\gamma$ be the function values at locations $\gamma$. The prior process can be expressed as

$$p(f) = p(f|\mathbf{f}_\gamma, \mathbf{f}_\beta)p(\mathbf{f}_\gamma|\mathbf{f}_\beta)p(\mathbf{f}_\beta), \tag{24}$$

where $p(\mathbf{f}_\beta)$ is defined as before, the prior conditional distribution of $\mathbf{f}_\gamma$ given $\mathbf{f}_\beta$ can be written as

$$p(\mathbf{f}_\gamma|\mathbf{f}_\beta) = \mathcal{N}(\mathbf{K}_{\gamma,\beta}\mathbf{K}_\beta^{-1}\mathbf{f}_\beta, \quad \mathbf{K}_\gamma - \mathbf{K}_{\gamma,\beta}\mathbf{K}_\beta^{-1}\mathbf{K}_{\beta,\gamma}),$$

and $p(f|\mathbf{f}_\gamma, \mathbf{f}_\beta)$ the prior conditional process conditioned on $\mathbf{f}_\gamma$ and $\mathbf{f}_\beta$ is a GP, which we denote as $\mathcal{GP}(m_{\mathbf{f}_\gamma, \mathbf{f}_\beta}, s_{\mathbf{f}_\gamma, \mathbf{f}_\beta})$ and has the following mean and covariance functions

$$m_{\mathbf{f}_\gamma, \mathbf{f}_\beta}(x) = [\mathbf{k}_{x,\gamma} \quad \mathbf{k}_{x,\beta}] \begin{bmatrix} \mathbf{K}_\gamma & \mathbf{K}_{\gamma,\beta} \\ \mathbf{K}_{\beta,\gamma} & \mathbf{K}_\beta \end{bmatrix}^{-1} \begin{bmatrix} \mathbf{f}_\gamma \\ \mathbf{f}_\beta \end{bmatrix} \tag{25}$$

$$s_{\mathbf{f}_\gamma, \mathbf{f}_\beta}(x, x') = k(x, x') - [\mathbf{k}_{x,\gamma} \quad \mathbf{k}_{x,\beta}] \begin{bmatrix} \mathbf{K}_\gamma & \mathbf{K}_{\gamma,\beta} \\ \mathbf{K}_{\beta,\gamma} & \mathbf{K}_\beta \end{bmatrix}^{-1} \begin{bmatrix} \mathbf{k}_{\gamma,x} \\ \mathbf{k}_{\beta,x} \end{bmatrix} \tag{26}$$

Following the same idea of Titsias [32], we consider a variational posterior written as

$$q(f) = p(f|\mathbf{f}_\gamma, \mathbf{f}_\beta)q(\mathbf{f}_\gamma, \mathbf{f}_\beta). \tag{27}$$

Now we show how to parameterize $q(\mathbf{f}_\gamma, \mathbf{f}_\beta)$ so that (27) defines an orthogonally decoupled GP. Note that if we parameterized this distribution as a full-rank Gaussian with no further restriction, it would be equivalent to just absorbing $\gamma$ into $\beta$ and would incur the computational complexity that we seek to avoid.

To obtain an orthogonally decoupled posterior, we use the form

$$q(\mathbf{f}_\gamma, \mathbf{f}_\beta) = q(\mathbf{f}_\gamma|\mathbf{f}_\beta)q(\mathbf{f}_\beta), \tag{28}$$

where $q(\mathbf{f}_\beta) = \mathcal{N}(\mathbf{m}_\beta, \mathbf{S}_\beta)$, and we define

$$q(\mathbf{f}_\gamma|\mathbf{f}_\beta) = \mathcal{N}(\mathbf{m}_{\gamma\perp\beta} + \mathbf{K}_{\gamma,\beta}\mathbf{K}_\beta^{-1}\mathbf{f}_\beta, \quad \mathbf{K}_\gamma - \mathbf{K}_{\gamma,\beta}\mathbf{K}_\beta^{-1}\mathbf{K}_{\beta,\gamma}) \tag{29}$$

for some variational parameter $\mathbf{m}_{\gamma\perp\beta}$. Note that $q(\mathbf{f}_\gamma|\mathbf{f}_\beta)$ is a Gaussian distribution that matches $p(\mathbf{f}_\gamma|\mathbf{f}_\beta)$ in (24) in covariance, but does *not* match in the mean unless $\mathbf{m}_{\gamma\perp\beta} = 0$. If we were to set $q(\mathbf{f}_\gamma|\mathbf{f}_\beta) = p(\mathbf{f}_\gamma|\mathbf{f}_\beta)$ we would recover the standard result using $\beta$ alone. This is because we would have effectively absorbed $\mathbf{f}_\gamma$ into the prior conditional process.

Since our choice for $q(\mathbf{f}_\gamma|\mathbf{f}_\beta)$ matches the prior in the covariance and has the same linear dependency on $\mathbf{f}_\beta$, the posterior process of $q(f)$ in (27) has a covariance function as (26). To find its mean function, let us first write $q(\mathbf{f}_\gamma, \mathbf{f}_\beta)$ as a joint distribution:

$$q\left(\begin{bmatrix}\mathbf{f}_\gamma\\\mathbf{f}_\beta\end{bmatrix}\right) = \mathcal{N}\left(\begin{bmatrix}\mathbf{m}_{\gamma\perp\beta} + \mathbf{K}_{\gamma,\beta}\mathbf{K}_\beta^{-1}\mathbf{m}_\beta\\\mathbf{m}_\beta\end{bmatrix}, \begin{bmatrix}\mathbf{K}_\gamma + \mathbf{K}_{\gamma,\beta}\mathbf{K}_\beta^{-1}(\mathbf{S}_\beta - \mathbf{K}_\beta)\mathbf{K}_\beta^{-1}\mathbf{K}_{\beta,\gamma} & \mathbf{K}_{\gamma,\beta}\mathbf{K}_\beta^{-1}\mathbf{S}_\beta\\\mathbf{S}_\beta\mathbf{K}_\beta^{-1}\mathbf{K}_{\beta,\gamma} & \mathbf{S}_\beta\end{bmatrix}\right)$$

We then can derive the posterior process mean function as

$$m(x) = \begin{bmatrix}\mathbf{k}_{x,\gamma} & \mathbf{k}_{x,\beta}\end{bmatrix}\begin{bmatrix}\mathbf{K}_\gamma & \mathbf{K}_{\gamma,\beta}\\\mathbf{K}_{\beta,\gamma} & \mathbf{K}_\beta\end{bmatrix}^{-1}\begin{bmatrix}\mathbf{m}_{\gamma\perp\beta} + \mathbf{K}_{\gamma,\beta}\mathbf{K}_\beta^{-1}\mathbf{m}_\beta\\\mathbf{m}_\beta\end{bmatrix} \tag{30}$$

To simplify the above expression, we write the inverse block matrix explicitly as

$$\begin{bmatrix}\mathbf{K}_\gamma & \mathbf{K}_{\gamma,\beta}\\\mathbf{K}_{\beta,\gamma} & \mathbf{K}_\beta\end{bmatrix}^{-1} = \begin{bmatrix}\mathbf{K}_{\gamma\perp\beta}^{-1} & -\mathbf{K}_{\gamma\perp\beta}^{-1}\mathbf{K}_{\gamma,\beta}\mathbf{K}_\beta^{-1}\\-\mathbf{K}_\beta^{-1}\mathbf{K}_{\beta,\gamma}\mathbf{K}_{\gamma\perp\beta}^{-1} & \mathbf{K}_\beta^{-1} + \mathbf{K}_\beta^{-1}\mathbf{K}_{\beta,\gamma}\mathbf{K}_{\gamma\perp\beta}^{-1}\mathbf{K}_{\gamma,\beta}\mathbf{K}_\beta^{-1}\end{bmatrix} \tag{31}$$

where we define

$$\mathbf{K}_{\gamma\perp\beta} = \mathbf{K}_\gamma - \mathbf{K}_{\beta,\gamma}\mathbf{K}_\beta^{-1}\mathbf{K}_{\beta,\gamma}. \tag{32}$$

After canceling several terms, we arrive at the expression

$$m(x) = \mathbf{k}_{x,\gamma}\mathbf{K}_{\gamma\perp\beta}^{-1}\mathbf{m}_{\gamma\perp\beta} - \mathbf{k}_{x,\beta}\mathbf{K}_{\beta,\gamma}\mathbf{K}_{\gamma\perp\beta}^{-1}\mathbf{m}_{\gamma\perp\beta} + \mathbf{K}_{x,\beta}\mathbf{K}_\beta^{-1}\mathbf{m}_\beta \tag{33}$$

A natural choice is to define $\mathbf{a}_\gamma = \mathbf{K}_{\gamma\perp\beta}^{-1}\mathbf{m}_{\gamma\perp\beta}$ (which agrees with the definition in Figure 1). In this case, we obtain

$$m(x) = (\mathbf{k}_{x,\gamma} - \mathbf{k}_{x,\beta}\mathbf{K}_{\beta,\gamma})\mathbf{a}_\gamma + \mathbf{K}_{x,\beta}\mathbf{K}_\beta^{-1}\mathbf{m}_\beta. \tag{34}$$

which is exactly the result for the orthogonally decoupled basis, as $\mathbf{m}_\beta = \mathbf{K}_\beta^{-1}\mathbf{a}_\beta$.

The decoupled basis can therefore be interpreted from the inducing perspective as a special case of a structured covariance. The key idea above is to use prior conditional matching, just as in the approach of Titsias [32], but to match only the covariance and not the mean.

## E  Expression for the Optimal Variational Parameters in Decoupled Bases

In the case of the Gaussian likelihood we can solve the variational inference problem 4 analytically, although doing so incurs a cost that scales cubically in $\alpha$ and prohibits the use of minibatches.

To make the results mirror the familiar expression for the optimal variational parameters in the coupled case [32], we use the basis

$$\mu = \Psi_\alpha\mathbf{a}, \qquad \Sigma = I + \Psi_\beta\mathbf{K}_\beta^{-1}(\mathbf{S} - \mathbf{K}_\beta)\mathbf{K}_\beta^{-1}\Psi_\beta^\top,$$

This basis is equivalent to the HYBRID and DECOUPLED bases through redefinition of parameters. The solution for $\mathbf{S}$ is exactly the same as in the coupled case:

$$\mathbf{S} = \left( \frac{1}{\sigma^2} \mathbf{K}_\beta^{-1} \mathbf{K}_{\beta X} \mathbf{K}_{X\beta} \mathbf{K}_\beta^{-1} + \mathbf{K}_\beta^{-1} \right)^{-1}$$

$$= \mathbf{K}_\beta \left( \frac{1}{\sigma^2} \mathbf{K}_{\beta X} \mathbf{K}_{X\beta} + \mathbf{K}_\beta \right)^{-1} \mathbf{K}_\beta .$$

For $\mathbf{a}$, we have

$$\mathbf{a} = \left( \frac{1}{\sigma^2} \mathbf{K}_{\alpha X} \mathbf{K}_{X\alpha} + \mathbf{K}_\alpha \right)^{-1} \mathbf{K}_{\alpha X} \mathbf{y}$$

## F  Experimental Details

In our experiments we use sensible defaults and do not hand tune for specific datasets. The full details are as follows:

**Kernel**    We use the sum of a Matern52 kernel with lengthscale $0.1\sqrt{D}$ and an RBF kernel with lengthscale $\sqrt{D}$, where $D$ is the input dimension. Both kernels are intialized to unit amplitude for regression and amplitude 5 for classification.

**Inducing point initalizations**    We use kmeans to initialize $\beta$ and use a random sample of the data for $\gamma$. We take care to use the same random seeds to ensure consistency between methods. For the DECOUPLED basis $\alpha$ we concatenate $\gamma$ and $\beta$ for a fair comparison with the other methods.

**Data preprocessing and splits**    The datasets we used had already been preprocessed to have zero mean and unit standard deviation. We construct test sets with a random 10% split. The splits are the same, so the results are directly comparable between our methods. The results from Klambauer et al. [18] used a different split from ours, however.

**Variational Parameter Initializations**    We initialize the variational parameters to the prior. I.e. zero mean and $\mathbf{S} = \mathbf{K}_\beta$ (NB the $\mathbf{B}$ in the DECOUPLED basis is initialized to near zero).

**Optimization**    We the adam optimizer with the default settings in the tensorflow implementation (including a learning rate of 0.001) for 20000 iterations. We use a step size of 0.005 for the natural gradient updates. For the non-conjugate likelihoods we increase from $10^{-5}$ to 0.005 linearly over the first 100 iterations, following the suggestion in Salimbeni et al. [29].

**Likelihood**    We initialize the Gaussian likelihood variance to 0.1.

**Minibatches**    We use a batch size of 1024 for data sub-sampling, and a batch of size 64 for the sub-sampling the columns of the $\mathbf{a}_\gamma^\top \mathbf{K}_\gamma \mathbf{a}_\gamma$ term in the ELBO.

We implemented all our methods in tensorflow, using on an open-source Gaussian process package, GPflow [21]. Our code [*] and datasets [†] are publicly available.

## G  Further Results

---

[*] https://github.com/hughsalimbeni/orth_decoupled_var_gps
[†] https://github.com/hughsalimbeni/bayesian_benchmarks

Table 3: Regression results normalized test likelihoods. High numbers are better. The coupled bases had $|\beta| = 400$ ($|\beta| = 300$ for the † bases), and the decoupled all had $\gamma = 700$, $\beta = 300$. We note that the orthogonal bases always outperform their coupled counterparts with the same $\beta$, but this does not hold for the DECOUPLED or HYBRID bases

| | N | D | COUPLED† | COUPLEDNAT† | COUPLED | COUPLEDNAT | ORTHNAT | ORTH | HYBRID | DECOUPLED |
|---|---|---|---|---|---|---|---|---|---|---|
| 3droad | 434874 | 3 | -0.7630 | -0.7632 | -0.7218 | -0.7228 | **-0.5947** | -0.6103 | -0.7617 | -0.9438 |
| houseelectric | 2049280 | 11 | 1.3130 | 1.3563 | 1.3383 | 1.3727 | **1.3899** | 1.3719 | 1.3092 | 0.6032 |
| slice | 53500 | 385 | 0.7816 | 0.7868 | 0.8321 | 0.8415 | **0.8776** | 0.8701 | 0.7852 | 0.0655 |
| elevators | 16599 | 18 | -0.4475 | -0.4455 | -0.4448 | **-0.4438** | -0.4479 | -0.4441 | -0.4585 | -0.4966 |
| bike | 17379 | 17 | 0.0059 | 0.0135 | 0.0321 | **0.0419** | 0.0271 | 0.0317 | -0.0318 | -0.1783 |
| keggdirected | 48827 | 20 | 1.0134 | 1.0158 | 1.0214 | 1.0223 | **1.0224** | 1.0216 | 1.0102 | 0.8947 |
| pol | 15000 | 26 | 0.0726 | 0.0821 | 0.1047 | 0.1132 | **0.1586** | 0.1451 | 0.0784 | -0.2502 |
| keggundirected | 63608 | 27 | 0.6984 | 0.6999 | 0.6994 | **0.7020** | 0.7007 | 0.6967 | 0.6878 | 0.6374 |
| protein | 45730 | 9 | -0.9531 | -0.9535 | -0.9375 | -0.9361 | **-0.9138** | -0.9165 | -0.9527 | -1.0464 |
| song | 515345 | 90 | -1.1902 | -1.1898 | -1.1890 | -1.1884 | **-1.1880** | -1.1882 | -1.1909 | -1.2266 |
| buzz | 583250 | 77 | -0.0566 | -0.0551 | -0.0512 | -0.0490 | **-0.0480** | -0.0484 | -0.0614 | -0.2285 |
| kin40k | 40000 | 8 | 0.0561 | 0.1580 | 0.2191 | **0.2234** | 0.1931 | 0.1777 | 0.1531 | -0.3877 |
| Mean | | | 0.0442 | 0.0588 | 0.0752 | 0.0814 | **0.0981** | 0.0923 | 0.0472 | -0.2131 |
| Median | | | 0.031 | 0.048 | 0.068 | 0.078 | **0.093** | 0.088 | 0.023 | -0.239 |
| Avg Rank | | | 2.917 | 4.000 | 5.417 | 6.750 | **7.083** | 6.250 | 2.583 | 1.000 |

Table 4: As Table 3 but reporting test RMSE. Lower numbers are better.

| | N | D | COUPLED† | COUPLEDNAT† | COUPLED | COUPLEDNAT | ORTHNAT | ORTH | HYBRID | DECOUPLED |
|---|---|---|---|---|---|---|---|---|---|---|
| 3droad | 434874 | 3 | 0.5166 | 0.5163 | 0.4946 | 0.4951 | **0.4332** | 0.4395 | 0.6179 | 0.5150 |
| houseelectric | 2049280 | 11 | 0.0639 | 0.0611 | 0.0615 | 0.0595 | **0.0583** | 0.0594 | 0.1286 | 0.0636 |
| slice | 53500 | 385 | 0.0840 | 0.0848 | 0.0787 | 0.0779 | **0.0730** | 0.0736 | 0.2112 | 0.0838 |
| elevators | 16599 | 18 | 0.3767 | 0.3760 | 0.3756 | 0.3753 | 0.3770 | **0.3752** | 0.3973 | 0.3812 |
| bike | 17379 | 17 | 0.2342 | 0.2324 | 0.2283 | **0.2261** | 0.2293 | 0.2282 | 0.2848 | 0.2438 |
| keggdirected | 48827 | 20 | 0.0883 | 0.0878 | 0.0874 | **0.0871** | 0.0871 | 0.0873 | 0.0980 | 0.0883 |
| pol | 15000 | 26 | 0.2073 | 0.2059 | 0.2012 | 0.2000 | **0.1906** | 0.1931 | 0.2867 | 0.2065 |
| keggundirected | 63608 | 27 | 0.1200 | 0.1196 | 0.1197 | **0.1191** | 0.1194 | 0.1202 | 0.1304 | 0.1223 |
| protein | 45730 | 9 | 0.6207 | 0.6216 | 0.6118 | 0.6113 | **0.5963** | 0.5972 | 0.6868 | 0.6209 |
| song | 515345 | 90 | 0.7954 | 0.7952 | 0.7944 | 0.7939 | **0.7936** | 0.7938 | 0.8275 | 0.7960 |
| buzz | 583250 | 77 | 0.2601 | 0.2606 | 0.2586 | 0.2579 | **0.2574** | 0.2576 | 0.3106 | 0.2617 |
| kin40k | 40000 | 8 | 0.2087 | 0.1885 | 0.1768 | 0.1746 | **0.1740** | 0.1776 | 0.3501 | 0.1887 |
| Mean | | | 0.2980 | 0.2958 | 0.2907 | 0.2898 | **0.2824** | 0.2836 | 0.3608 | 0.2977 |
| Median | | | 0.221 | 0.219 | 0.215 | 0.213 | **0.210** | 0.211 | 0.299 | 0.225 |
| Avg Rank | | | 6.083 | 5.167 | 3.750 | 2.417 | **1.833** | 2.500 | 8.000 | 6.250 |

Table 5: Classification accuracy results, including the results from Klambauer et al. [18].

| | N | D | K | Selu | COUPLED | COUPLEDNAT | ORTH | ORTHNAT | HYBRID | DECOUPLED |
|---|---|---|---|---|---|---|---|---|---|---|
| adult | 48842 | 15 | 2 | 84.76 | 85.85 | **86.11** | 85.65 | **86.15** | 85.73 | 84.37 |
| chess-krvk | 28056 | 7 | 18 | **88.05** | 67.38 | 60.23 | 67.76 | 60.70 | 59.34 | 53.56 |
| connect-4 | 67557 | 43 | 2 | **88.07** | 85.54 | 86.44 | 85.99 | 86.33 | 85.13 | 83.12 |
| letter | 20000 | 17 | 26 | **97.26** | 95.69 | 93.22 | 95.77 | 93.45 | 95.26 | 92.68 |
| magic | 19020 | 11 | 2 | 86.92 | 89.24 | **89.50** | 89.35 | **89.42** | 89.19 | 88.33 |
| miniboone | 130064 | 51 | 2 | 93.07 | 93.21 | **93.60** | 93.49 | **93.59** | 93.36 | 92.04 |
| mushroom | 8124 | 22 | 2 | **100.00** | **100.00** | **100.00** | **100.00** | **100.00** | **100.00** | **100.00** |
| nursery | 12960 | 9 | 5 | **99.78** | 97.30 | 97.30 | 97.30 | 97.30 | 97.30 | 97.29 |
| page-blocks | 5473 | 11 | 5 | 95.83 | **97.99** | 97.79 | 97.21 | 97.21 | 97.49 | 96.98 |
| pendigits | 10992 | 17 | 10 | 97.06 | **99.65** | 99.64 | **99.66** | 99.64 | **99.66** | 99.62 |
| ringnorm | 7400 | 21 | 2 | 97.51 | **98.92** | 98.78 | 98.78 | 98.78 | **98.86** | **98.92** |
| statlog-landsat | 6435 | 37 | 6 | 91.00 | 90.26 | **91.45** | 91.28 | 91.08 | **91.35** | 90.35 |
| statlog-shuttle | 58000 | 10 | 7 | **99.90** | 99.87 | 99.74 | **99.90** | **99.81** | 99.79 | 99.80 |
| thyroid | 7200 | 22 | 3 | 98.16 | 99.41 | **99.56** | **99.47** | 99.31 | **99.52** | 99.13 |
| twonorm | 7400 | 21 | 2 | **98.05** | 97.67 | 97.65 | 97.65 | 97.72 | 97.69 | 97.72 |
| wall-following | 5456 | 25 | 4 | 90.98 | 94.79 | 95.64 | 95.56 | **95.76** | 93.07 | 91.48 |
| waveform | 5000 | 22 | 3 | 84.80 | 85.80 | 86.54 | 86.13 | 86.21 | 86.53 | **87.55** |
| waveform-noise | 5000 | 41 | 3 | **86.08** | 82.59 | 82.71 | 82.93 | 83.12 | 83.05 | 82.71 |
| wine-quality-white | 4898 | 12 | 7 | **63.73** | 57.14 | 58.61 | 57.05 | 59.56 | 56.58 | 55.71 |
| Mean | | | | **91.6** | 90.4 | 90.2 | 90.6 | 90.3 | 89.9 | 89.0 |
| Median | | | | 93.1 | 94.8 | 93.6 | **95.6** | 93.6 | 93.4 | 92.0 |
| Avg Rank | | | | 4.16 | 3.89 | 3.53 | 3.68 | **3.42** | 3.89 | 5.42 |

(a) Test log-likelihood

(b) Test MAE

Figure 3: Test log-likelihood (a), and accuracy (b) for the large scale experiment. The ELBO is reported in the main text, Figure 2c

Table 6: As Table 5 but reporting test log-likelihoods. The test log-likelihood results from [18] were not reported

|  | N | D | K | COUPLED | COUPLEDNAT | ORTH | ORTHNAT | HYBRID | DECOUPLED |
|---|---|---|---|---|---|---|---|---|---|
| adult | 48842 | 15 | 2 | -0.3048 | **-0.2970** | -0.3045 | **-0.2973** | -0.3067 | -0.3234 |
| chess-krvk | 28056 | 7 | 18 | **-2.1239** | -3.2821 | -2.1625 | -3.2145 | -3.0443 | -3.5380 |
| connect-4 | 67557 | 43 | 2 | -0.3160 | **-0.3009** | -0.3086 | **-0.3017** | -0.3244 | -0.3680 |
| letter | 20000 | 17 | 26 | -0.2316 | -0.4892 | **-0.2276** | -0.4793 | -0.2810 | -0.4861 |
| magic | 19020 | 11 | 2 | -0.2666 | **-0.2641** | -0.2658 | **-0.2646** | -0.2697 | -0.2863 |
| miniboone | 130064 | 51 | 2 | -0.1680 | **-0.1584** | -0.1618 | **-0.1585** | -0.1645 | -0.1902 |
| mushroom | 8124 | 22 | 2 | **-0.0006** | **-0.0007** | **-0.0009** | **-0.0008** | **-0.0007** | **-0.0007** |
| nursery | 12960 | 9 | 5 | **-0.2228** | -0.2233 | **-0.2225** | -0.2229 | -0.2236 | -0.2241 |
| page-blocks | 5473 | 11 | 5 | -0.1301 | **-0.0989** | -0.1328 | -0.1112 | -0.1368 | -0.1538 |
| pendigits | 10992 | 17 | 10 | -0.0251 | **-0.0216** | -0.0209 | **-0.0207** | **-0.0216** | -0.0241 |
| ringnorm | 7400 | 21 | 2 | **-0.0345** | -0.0458 | -0.0466 | -0.0465 | -0.0410 | -0.0418 |
| statlog-landsat | 6435 | 37 | 6 | -0.4503 | -0.4102 | -0.3956 | -0.3920 | **-0.3771** | -0.4938 |
| statlog-shuttle | 58000 | 10 | 7 | **-0.0047** | -0.0199 | **-0.0049** | -0.0174 | -0.0170 | -0.0166 |
| thyroid | 7200 | 22 | 3 | -0.0257 | -0.0133 | **-0.0115** | -0.0211 | -0.0127 | -0.0290 |
| twonorm | 7400 | 21 | 2 | **-0.0590** | -0.0588 | **-0.0590** | -0.0595 | -0.0607 | -0.0620 |
| wall-following | 5456 | 25 | 4 | -0.2032 | -0.1674 | **-0.1514** | -0.1537 | -0.3191 | -0.4102 |
| waveform | 5000 | 22 | 3 | -0.6207 | -0.5572 | -0.5640 | **-0.5038** | -0.5160 | -0.5469 |
| waveform-noise | 5000 | 41 | 3 | -0.7650 | -0.7416 | -0.7096 | **-0.6778** | -0.6893 | -0.7673 |
| wine-quality-white | 4898 | 12 | 7 | -2.8884 | -2.5400 | -2.5681 | **-2.5069** | -2.7557 | -2.7921 |
| Mean |  |  |  | -0.4653 | -0.5100 | **-0.4378** | -0.4974 | -0.5033 | -0.5660 |
| Median |  |  |  | -0.223 | -0.223 | **-0.222** | -0.223 | -0.270 | -0.286 |
| Avg Rank |  |  |  | 3.553 | 3.026 | 2.868 | **2.737** | 3.605 | 5.211 |

## Footnotes

*We follow the conventional abuse of notation by writing the process as if it has a density. See Matthews [22] for a rigorous treatment that defines the posterior processes as in terms of Radon-Nikodym derivative with respect to the prior.