[Reviews · NeurIPS 2018]

Reviewer 1



#After after the author's rebuttal: Thank you for your feedback. I agree the standard error can be misleading due to the correlations induced by the datasets while the standard error of the ranks is more informative. After the discussion with other reviewers, I think adding the motivation (or derivation) for Equation 9 will definitely make the paper stronger and I encourage the authors to do so in their final version. This paper proposes a novel RKHS parameterization of decoupled GPs that admits efficient natural gradient computation. Specifically, they decompose the mean parameterization into a part that shares the basis with the covariances, and an orthogonal part that models the residues that the standard decoupled GP (Chen and Boots, 2017) fails to capture. This construction allows for a straightforward natural gradient update rule. They show this algorithm demonstrates significantly faster convergence in multiple experiments. Although the computational time is larger than the Chen and Boot’s approach, they show that the additional computation is worthwhile for the substantial performance improvement. Clarity: This paper is well-written. Originality: The decomposition of orthogonally decoupled basis and residual of the mean and covariance parameterization is entirely novel to my knowledge. Significant: This orthogonally decoupled approach is highly significant in advancing the field of scalable GP inference. The algorithm is extensively tested against benchmark datasets (both regression and classification) and demonstrated the orthogonally decoupled basis is superior to the coupled basis. Other comments/suggestions: - In the abstract session, the word “coupled” is kind of confusing since the proposed method along with the standard approach are both called “decoupled” (line 7 and 12). I notice the word “coupled” appears later in the text (line 69) - In the experiment session, when comparing the wall-clock for different algorithms, maybe the authors should provide the standard errors of the different measurements over all the testing data, so that we can see whether the log likelihood are significantly from each other and whether the ORTHNAT significantly outperforms others. Also for Figure 2, “EXACTCOUPLED” and “EXACTDECOUPLED” is not specified in the texts. It might also be better to change the descriptions of the three panels to be in the (a), (b) and (c) order, or change the order of the panels. - line 285: “Table 6” -> “Table 6 (Appendix)” since Table 6 is in the supplementary materials Typos: - line 106: “to searching” -> “to search” - line 277: “The the” -> “The”

Reviewer 2



## [Updated after author feedback] Thank you for your feedback. Your discussion about the uncertainties on the experiments makes sense and you have a good point. Standard deviations on the mean and median would indeed not make much sense. I would still have preferred if you had performed a cross-validation for each dataset-method pair in Appendix F, but I acknowledge that this will be computationally very demanding. Thank you for including a discussion of Eq. (9). It is insightful and I hope you will include it in the paper. It does, however, not provide the motivation I was hoping for and it is still not clear to me how you came up with the proposed basis in the first place. The choice clearly works, but I see the lack of motivation as the weakest point of the paper. Including a discussion on how you found the particular form of the basis will make the paper significantly stronger and it might spark further developments. I genuinely hope you will add it. Based on the reviewer discussions, I have decided to reduce my score from 9 to 8. The reason is alone the missing motivation for the proposed basis. The origin of the key contribution of the paper is a mystery, and it should not be so. Still, I strongly recommend an accept of the paper. ## Summary The paper presents an extension to the Decoupled Gaussian Processes formulation of Cheng & Boots (2017), which addresses optimisation difficulties with the original method. The authors propose a new decoupled basis for the mean and covariance that extends the original basis with an orthogonal part, capturing the residues missed by the original method. The authors go on to show that the proposed basis leads to the existence of natural parameters and derive gradient update rules for these. This allows for using natural gradient descent to optimise the model, which is shown to outperform the original method, both in terms of convergence rate and performance on regression and classification datasets. ## Quality The paper is of high technical quality, with adequate details in the derivations to follow them. It is very good to see that the performance has been evaluated on many (12) datasets. However, I urge you to also perform a cross-validation and report mean and standard deviation across the folds. At the very least, you should report the spread of the performance metrics across the datasets in table 1 and 2. The values are all very similar, and it is difficult to judge whether your method is significantly better. It is a shame that such an otherwise strong paper misses these key metrics. Since the proposed method can represent the mean and variance with different amounts of inducing points, it would be interesting to see the experiments evaluated with a metric that also uses the variance, e.g. NLPD. ## Clarity The paper is very clearly written with an excellent structure. The idea and story are easy to follow, and motivations and comparisons to related work are provided along the way. I very much enjoyed reading it. Some motivation for the exact form of Eq. (9) would, however, be nice to see. From what follows, it is clear that it works, but how did you come up with it in the first place? Such motivation would be valuable to readers at the beginning of their research careers. ## Originality As the proposed methods is an extension of a previous method it is, by nature, somewhat incremental. However, the proposed new, decoupled basis and the following derivations of natural parameters and natural gradient descent update rules makes for a solid and novel work. ## Significance The paper provides a solid and significant extension to a recently proposed method. The work is of high quality and a significant contribution to the field of scalable GP methods. ## Minor comments Line 285: Table 6 -> Table 1

Reviewer 3



Edit 2: While reviewing the paper and discussing with other reviewers, I reframed the approximation of Cheng and Boots as a structured variational approximation in the Titsias's framework like this: https://imgur.com/a/E6uUKXh, and similarly for the method proposed in this paper: https://imgur.com/a/tbx1fMs. While these might seem trivial in hindsight and that they didn't bring much intuition, I think these could be useful in making connections between methods and previously published results. I hope the authors find these useful. Edit after the rebuttal period 1: The author response has sufficiently addressed my concerns, though I would still want to understand how the approximation was discovered/derived. I also think that it could be good, as a future research direction, to relate the approximation of Cheng and Boots and the approximation that this paper is using to the Titsias' framework without the need of RKHS, and the natural gradient version of Hensman et al. Summary: This paper introduces an orthogonally decoupled sparse variational approximation for Gaussian process regression and classification models, extending the decoupled framework of Cheng and Boots (2017). The idea is that a combination of a particular partitioning of the inducing points and a specific parameterisation of the mean and covariance functions lead to an efficient sparse posterior such that natural gradients can be obtained cheaply, enabling natural gradient methods to be used to optimise the variational lower bound. Experiments on multiple regression and classification datasets demonstrate the performance of the new decoupled approach and its version using natural gradients. Comments: 1. the parameterisation of the mean and covariance in equations 9 is particularly interesting, as it allows useful cancellations to happen when moving to equations 10. 2. I'm not sure how this particular parameterisation pops up in the first place, could the thought process be elaborated -- I think this could be useful for deriving further approximations based on this decoupled framework? for example, the natural gradient version of Cheng and Boots as in appendix C2 was considered first, and then somehow you realised that inserting (I - \phi_\beta K_\beta^{-1} \phi_\beta) in front of \phi_gamma gives decoupled natural gradients. This does not look trivial. 3. just an observation: the hybrid approach seems to perform poorly compared to the natural gradient version of Orth and Coupled, but seems to be better than Orth. 4. I am particularly intrigued that the natural version of the coupled approach -- which I think is the natural gradient approach of Hensman et al -- performs *much better* than the decoupled approach of Cheng and Boots (2017). However, Cheng and Boots compared their decoupled approach to SVI [I thought this is the approach of Hensman et al], and the decoupled approach outperforms SVI in all experiments they considered. Have you looked at this contradictory results in detail? Or I'm missing something here? 5. While I think the experimental results, especially on the regression dataset, are pretty solid, I'm wondering what is the fairest comparison protocol here? Previous sparse GP works basically assumed the same number of pseudo points for all methods and looked at the predictive performance after a number of iterations [provided that all methods have the same computational complexity]. The decoupled approaches have two set of pseudo points, and the complexity is a little bit different compared to previous approaches. The deltas between methods in the experimental tables are also very small to concretely say one method is better than the other, given they all have slightly different complexities and hence running times. 6. Making some of the ranking results in table 2 bold is, in my opinion, a bit misleading, as the methods perform very similarly and the std error for the mean rank seems to be very large here. 7. The paper is generally well-written, but I found it's hard to understand what the approximation is intuitively doing. Perhaps some visualisation of the methods [coupled vs decoupled vs orthogonally decoupled] on some 1D, 2D tasks could be useful, i.e. what the pseudo-points are doing, what happens when you optimise them, will the methods get to exact GP when the number of pseudo points = number of training points, does the FITC pathology described in Bauer et al (2016) exist here, ...